# The Principle of Steam Explosion Technology and Its Application in Food Processing By-Products

**DOI:** 10.3390/foods12173307

**Published:** 2023-09-02

**Authors:** Changrong Wang, Mengfan Lin, Qingyu Yang, Chenying Fu, Zebin Guo

**Affiliations:** 1College of Food Science, Fujian Agriculture and Forestry University, Fuzhou 350002, China; wcr9948@163.com (C.W.); lin_mengfan@163.com (M.L.); qingyu980206@163.com (Q.Y.); fcy7600@163.com (C.F.); 2Integrated Scientific Research Base of Edible Fungi Processing and Comprehensive Utilization Technology, Ministry of Agriculture and Rural Affairs, Fuzhou 350002, China

**Keywords:** steam explosion, principle, application

## Abstract

Steam explosion technology is an emerging pretreatment method that has shown great promise for food processing due to its ability to efficiently destroy the natural barrier structure of materials. This narrative review summarizes the principle of steam explosion technology, its similarities and differences with traditional screw extrusion technology, and the factors that affect the technology. In addition, we reviewed the applications in food processing by-products in recent years. The results of the current study indicate that moderate steam explosion treatment can improve the quality and extraction rate of the target products. Finally, we provided an outlook on the development of steam explosion technology with a reference for a wider application of this technology in the food processing field.

## 1. Introduction

As the global population continues to grow, there is a need to utilize our limited resources to meet the growing demand for food. The Food and Agriculture Organization reports that approximately 1.6 billion tons of food is wasted globally each year, accounting for approximately one-third of total food production [1]. Out of the 1.5 billion tons of fruits and vegetables produced each year, approximately 500 million tons are wasted or transformed into by-products [2]. Meat-producing animals generate 40% of by-products, and 17 million tons of animal by-products are produced annually in Europe alone [3]. These food processing by-products have a hard natural barrier structure that makes them difficult to process [4]. As a result, most of food processing by-products are incinerated or disposed as waste, which contributes to environmental pollution [5]. Studies have shown that food processing by-products are rich in nutrients and bioactive compounds such as polysaccharides, polyphenols, dietary fibers, and flavonoids [6], which play important roles in human health and have great potential as functional food ingredients. To reduce food waste and improve food utilization, it is necessary to seek suitable pretreatment methods to open up the internal structure, change the chemical composition, and improve the quality of by-products [7] for better utilization of these resources.

Currently, common food pretreatment methods include physical, chemical, and biological methods [8]. The chemical method mainly destroys the structure of raw materials by adding acid or alkali to achieve the purpose of pretreatment [9]. The chemical method has the advantage of short treatment time, but there are problems such as waste liquid pollution of the environment and the possible introduction of toxic and harmful chemical groups during the treatment process [10], so there are certain risks when applied to food processing. Biological methods mainly use enzymes or microorganisms to enzymatically dissolve or ferment raw materials, thus completing the pretreatment of raw materials [11]; the reaction of the method is mild, but the reaction conditions are extremely harsh and expensive, and it is difficult to realize the industrialization of the application [4]. Compared to chemical and biological methods, physical methods have received extensive attention from scholars due to their advantages such as high efficiency, low cost, and absence of pollutants [12].

Steam explosion technology stands out among many physical pretreatment methods due to its obvious advantages in energy consumption and treatment efficiency [13]. The technology was first proposed by W. H. Mason and applied to the manufacture of fiber compression boards [14]. Steam explosion is no longer limited to areas such as pulping and wood processing [15], animal feed processing [16], and bioenergy production [17], but to food processing as well. For instance, Wang et al. [18] reported that the content of bioactive components in rapeseed oil improved with steam explosion, and Yi et al. [19] concluded that *Achyranthis bidentatae* had higher polysaccharide content following steam explosion. Steam explosion can be used not only on its own but also in combination with other pretreatment methods. Xi et al. [20], who used steam explosion pretreatment coupled with ultrasonic extraction, acid extraction, alkaline extraction, and enzymatic extraction, found that these combined methods increased the yield of soluble dietary fiber (SDF) from highland barley. Therefore, steam explosion technology has unique advantages in food processing due to its excellent processing effects on materials and has now become a topic of significant interest in food science research.

This narrative review summarizes the principle and applications of steam explosion technology. Additionally, we review the limitations of steam explosion technology and provide an outlook on its development prospect in the food field, aiming to provide some reference for the further application of steam explosion technology in the food processing by-products.

## 2. Principle and Process of Steam Explosion Technology

In steam explosion technology, the raw material is placed in a confined environment at high temperature (160 to 260 °C) and pressure (0.69 to 4.83 MPa) [18]. The raw material is moistened by superheated saturated water vapor. The high pressure makes the water vapor fill the cell tissue interstices, and after a period of time, the high pressure is released into the atmosphere within milliseconds (≤0.00875 s). Thermal energy is converted into mechanical energy [21], so that the saturated water vapor in the tissue cells and high temperature liquid water adiabatic expansion “burst” and the surface structure of the cell wall ruptures into a microporous shape [22], allowing the release of small molecular weight substances from the cells while completing the separation of components and structural changes of raw materials [23].

Steam explosion consists of two stages: (1) vapocracking and (2) explosive decompression [24]. In vapocracking, similar to a thermochemical reaction (a steam explosion equipment installation diagram can be seen in Figure 1a), the saturated water steam generated by the steam generator fills the reaction chamber, making the confined reaction chamber a high-temperature and high-pressure environment. Under high pressure, the water vapor quickly fills the material inside and penetrates the tissue cell internal space, so that the intracellular water temperature rises. Under high-temperature steam, hemicellulose and lignin undergo different degrees of hydrolysis [25], which destroys the physical and chemical bonds between hemicellulose, cellulose, and lignin, thereby reducing the adhesive strength and facilitating the mechanical separation in the subsequent explosive decompression stage [26]. In explosive decompression, when the reactor’s inner cavity is connected to the external environment through a piston (the piston system works as depicted in Figure 1b), the high pressure in the system is released into the atmospheric pressure within milliseconds, and the saturated water vapor and vaporized high-temperature liquid water inside the cell undergo adiabatic expansion, destroying the cell wall of the material [27]. The release of physical energy is completed by the expanding gas in the form of shock waves. The internal structure of the material changes, and the mechanical forces lead to the destruction of chemical bonds in cellulose and to structural rearrangement, making it easy to change the cellulose into an ordered structure [7]. In this process, some active substances in the material can be effectively separated [28].

## 3. Comparison Steam Explosion and Traditional Screw Extrusion

Steam explosion and traditional extrusion expansion are similar heat treatment technologies. In both technologies, the material is placed in a high-temperature and high-pressure confined environment [29]. High pressure in the system is released instantaneously, and the water vapor present in the pores of the material expands, causing physicochemical changes inside the material [30].

### 3.1. Effect of Steam Explosion and Traditional Screw Extrusion on Material Treatment

Steam explosion has similar effects to traditional conventional screw extrusion. Both technologies modify the material in a way that minimizes food degradation and destroys anti-nutrients [31]. In addition, the microstructure and physicochemical properties of the material can be modified by steam explosion and traditional screw extrusion with improvements in the functional properties [32]. Wang et al. [33] and Qiao et al. [34], who subjected sweet potato residue to steam explosion and screw extrusion, respectively, reported that sweet potato residue SDF became more loose and porous; that the extracted amount of SDF increased; and that the water-holding capacity, oil-holding capacity, and swelling capacity improved to different degrees. Furthermore, Wang et al. [35] subjected wheat bran to extrusion and found that wheat bran SDF was enhanced along with its antioxidant capacity, similar to the findings of Feng et al. [36], who subjected wheat bran to steam explosion.

### 3.2. Differences and Similarities between Steam Explosion and Traditional Screw Extrusion

Compared with traditional screw extrusion, steam explosion has the following advantages. First, high adaptability: there are no requirements on the form or composition of raw materials [37]. Second, high explosion efficiency: the explosion is completed within 0.00875 s. There is a short working time, high power, and concentrated energy. Third, adequate processing results: in traditional screw extrusion, the extrusion process of raw materials is poor, and the material temperature is difficult to control [38]. Fourth, significant economic advantages: the traditional screw extrusion relies on the screw to process the material, which causes high wear and tear on the screw and requires regular replacement [31]. At the same time, energy consumption is a major part of the operating costs of steam explosion equipment, while steam explosion technology has a very high energy conversion efficiency (thermal energy to mechanical energy), so that the actual energy consumption is at a very low level [26]. Table 1 shows the difference between steam explosion and traditional screw extrusion. Steam explosion technology, due to its high efficiency, high adaptability, and economic characteristics, has broader application prospects.

## 4. Factors Influencing the Effectiveness of Steam Explosion

The effect of steam explosion on materials is affected by the equipment (external factors) and material (internal factors). Specifically, the factors include steam explosion pressure, residence time, loading ratio, pressure release duration, material characteristics (type, moisture content, tissue structure, morphology, and particle size), and presoaking treatment. Among them, steam explosion pressure and residence time are the most important factors affecting steam explosion [39].

### 4.1. Steam Explosion Pressure and Residence Time

Steam explosion pressure and residence time can be regulated based on the material. Hard materials such as bovine bones need to be treated with higher steam explosion pressure and longer residence time to effectively destroy the dense and orderly network of bovine bones and make the bones brittle [24]. In contrast, lower steam explosion pressure and shorter residence time are required to break the wall of sweet potato pomace and improve the yield of sweet potato pomace SDF [33]. Researchers commonly use the severity factor (*logR*_0_) to express the combined effects of steam burst temperature and residence time [40], which can be expressed by the following equation:(1)$${logR}_{0}=log(t\times e^{\frac{T_{r}-T_{b}}{14.75}})$$
where *t* is the residence time (min); *T_r_* is the steam temperature during treatment (°C); and *T_b_* is the base temperature (°C), generally 100 °C.

The above equation is applicable to the vapocracking stage. Yu et al. [26] argued that the very short time in the explosive decompression process can be regarded as adiabatic expansion and proposed to express the work performed by the adiabatic expansion in the explosive decompression process in terms of explosive power density (EPD, J/s·m^3^) to complement the above equation, which can be expressed by the following equation:(2)$$P_{epd}=\frac{{\Delta H}_{1}+{\Delta H}_{s}+{\Delta H}_{m}}{t\times V}$$
where ΔH_1_ is the enthalpy difference of liquid water (J); ΔH*_s_* is the enthalpy difference of saturated steam (J); ΔH*_m_* is the enthalpy difference of material (J); *t* is the expansion time (s); and V is the volume of steam explosion reactor (m^3^).

Explosion time and the volume of the steam explosion reactor are inherent parameters of the steam explosion equipment and are usually determined by its own construction [7]. Based on Equation (2), at the same volume of the steam explosion vessel, when the enthalpy difference is larger, the power generated by explosion is larger, the efficiency of conversion of thermal energy into mechanical energy is higher, and the steam explosion fractality factor and explosion power density together reflect the whole process of steam explosion.

### 4.2. Presoaking Treatment

Presoaking before steam explosion affects steam explosion by increasing the moisture content of the material. The main purpose of presoaking is to soften and moisten the fiber and hence reduce mechanical damage. During presoaking, the free water content within the cell increases significantly. In the vapocracking stage, the superheated saturated water vapor converts the water within the cell into high-temperature liquid water, which contributes to the subsequent explosive decompression stage of the energy conversion to improve the processing efficiency [41]. To improve the effectiveness of steam explosion, researchers have presoaked the material with an acid or alkaline solution. Wang et al. [42], who presoaked orange peel with dilute sulfuric acid prior to steam explosion, found that the content and physicochemical and functional properties of orange peel SDF were enhanced. Sorensen et al. [43] reported that dilute sulfuric acid during presoaking resulted in greater sugar yields after steam explosion than the non-presoaked biomass. M. A. F. et al. [44] found a significant increase in the amount of cellulose extracted from empty fruit bunches after steam explosion using NaOH as the presoaking solution.

### 4.3. Form of Material

Particle size, porosity, and morphological structure of the material affect the heat transfer of steam to the material, thereby affecting the effectiveness of steam explosion [45]. Studies have shown that the size of particle size is correlated with the effectiveness of steam explosion. Liang et al. [46] reported that when apple pomace was treated with steam explosion, apple pomace SDF yield increased rapidly at ≥20 mesh. The highest yield was obtained at 64.72 mesh and decreased at >100 mesh. This is because the raw material particle size becomes small, the surface area involved in the steam explosion reaction increases, and thus the treatment is enhanced. However, too small a particle size in the steam explosion treatment is prone to agglomeration, which prevents the further penetration of saturated steam, which in turn makes the steam explosion reaction is not sufficient [39]. Therefore, the particle size of the material can be reduced by grinding before steam explosion to maximize the steam explosion effect.

## 5. Application of Steam Explosion Technology in Food Processing By-Products

Steam explosion technology uses water vapor to treat the raw material and therefore does not produce any toxic or hazardous substances [13], and at the same time, because of its excellent processing effect and applicability, it has been widely researched in the field of food processing, including wall-breaking treatment of raw materials, extraction of bioactive components, and improvement of functional properties of bioactive components. Figure 2 presents an overview of the application of steam explosion technology in food processing by-products.

### 5.1. Application in Animal-Based Processing By-Products

Most animal-based processing by-products are resistant to degradation and require pretreatment with chemical reagents, such as sulfuric acid, hydrochloric acid, and sodium hydroxide [47]. This pretreatment method requires long reaction times and generates large amounts of waste liquid that pollute the environment. Considering the social and economic benefits, researchers have used steam explosion, an efficient and environmentally friendly technology, for the pretreatment of animal-based processing by-products (Table 2).

Steam explosion can approximate the adiabatic expansion process in the explosive decompression stage, which can complete the conversion from thermal energy to mechanical energy in a very short time, so the shear force generated by the high conversion efficiency can easily cope with the hard material. Steam explosion has an adequate wall-breaking effect, which facilitates the extraction of target products for subsequent extraction. Scopel et al. [48] used steam explosion as a pretreatment method for alkali extraction of waste leather gelatine, which tripled the gelatine extraction rate while obtaining high-quality gelatine. Ngasotter et al. [49] used steam explosion to prepare chitin nanocrystals from shrimp shell chitin and found that steam explosion is faster and requires a lower acid concentration and amount compared to acid hydrolysis. The reason for the increased protein extraction rate by steam explosion treatment may be that steam explosion alters the subunits of proteins, leading to depolymerization and a wider distribution of molecular weights, resulting in the formation of non-disulfide bonds between protein molecules [50]. The reason why steam explosion treatment improves the solubility of denatured proteins is that during the steam explosion process, hydrophilic groups interact with proteins, leading to Maillard reactions [51]. In addition, high solubility proteins indicate the ability to release more natural peptides [52], which means that steam explosion treatment may improve certain functional properties of proteins [53].

**Table 2 foods-12-03307-t002:** Application of steam explosion technology to animal-based processing by-products.

By-Products	Extract	Steam Explosion Conditions	Results	References
Bovine hide	Gelatin	Steam explosion temperature: 110, 120, and 130 °C Residence time: 60, 300, 600 s.	The increase in temperature and time significantly influenced the obtainment of gelatins with the highest dry matter, protein, viscosity, molecular weight, and amino acid profile, as well as the lowest conductivity, ash content, and pH levels.	[54]
Cattle bone	Collagen peptides	Steam explosion pressure: 2.0 MPa Residence time: 30 min.	The recovery rate of protein reached 60.5%. Destroyed the physical and chemical structures of cattle bone.	[24]
Tuna bone powder	Calcium	Steam explosion pressure: 0.6 MPa Residence time: 300 s.	Effectively shortened the time of the bone powder preparation process, and the obtained tuna bone powder had a smaller particle size and higher product quality.	[55]
Fish backbone	Protein	Steam explosion temperature: 159 °C Residence time: 120 s.	Both nitrogen recovery and free amino acids of the hydrolysates increased. Stronger free radical scavenging activity.	[56]
Chicken bone	Protein	Steam explosion temperature: 121 °C Residence time: 90 min.	The content of soluble protein and protein digestibility in chicken bone powder were significantly increased.	[57]
Bovine bone	Protein and minerals	Steam explosion pressure: 0.5, 1.0, 1.5, 2.0, and 2.5 MPa Residence time: 5, 10, 15, 20, 25, and 30 min.	Effectively promoted mineral dissolution and protein degradation while reduced energy consumption. Higher mineral release and protein digestibility during GI digestion while revealing no obvious cytotoxicity.	[58]
Chicken feathers	Keratin	Steam explosion pressure: 1.4, 1.6, 1.8, and 2.0 MPa, Residence time: 160 s.	Dramatically increasing the extraction and dissolubility of feather keratins in polar solvents such water, salt solution, and weak bases, as well as enzymatic accessibility.	[59]
Duck feathers	Keratin	Steam explosion pressure: 1.6 MPa Residence time: 1 min.	The extraction rate of feathers and the yield of keratin were improved.	[60]
Porcine hoof shells	Peptone	Steam explosion pressure: 0.5, 1.0, 1.5, 2.0, and 2.3 MPa Residence time: 5, 10, 15, 20, 25, and 30 min.	Porcine hoof shells can a peptone substitute for fermentation culture.	[47]
Shrimp shell	Chitin	Steam explosion temperature: 121 °C Residence time: 23.7 min.	Compared to conventional acid hydrolysis, the steam explosion method was faster and required a lesser acid concentration and quantity. The crystallinity index was higher and could be decomposed at lower temperatures.	[49]
Shrimp shell	Chitin	Steam explosion temperature: 179 °C Residence time: 7, 9, and 12 min.	Density and crystallinity index became lower; porosity and swelling become larger.	[61]
Chicken sternal cartilage	Chondroitin sulfate	Steam explosion pressure: 1.0–1.6 MPa Residence time: 60–140 s.	Highest recovery and total yield increased.	[62]

### 5.2. Application in Plant-Based Processing By-Products

Researchers have applied steam explosion technology more extensively in plant-based raw materials than in animal-based raw materials [47]. This phenomenon is mainly attributed to the characteristics of steam explosion technology. The shear force generated by steam explosion has an excellent wall-breaking effect, which can cause damage to dense plant tissues and hard cell wall structures, opening the natural anti-extraction barrier of plants, facilitating solute-solvent accessibility and internal mass transfer during the extraction process, and promoting the extraction of active ingredients [63]. Also, steam explosion promotes the degradation of polymers into biologically active substances. For example, the reason for the increased SDF extraction rate after steam blasting treatment is that steam blasting destroys the structure of the feedstock, and degradation of cellulose and hemicellulose occurs, thus promoting SDF release and extraction [64]. Therefore, steam explosion technology is highly suitable for plant-based raw materials. Table 3 shows that the extraction of active compounds, e.g., polyphenols, flavonoids, and SDF, increased under optimum steam explosion conditions.

### 5.3. Effect of Steam Explosion on Bioactive Substances in Food Processing By-Products

#### 5.3.1. Effect of Steam Explosion on Structural Properties of Bioactive Components

##### Effect of Steam Explosion on the Apparent Morphology of Bioactive Components

Steam explosion increases the extraction rate of active ingredients in the material and has an impact on its structural properties. This phenomenon occurs as a result of acid hydrolysis, thermal degradation, mechanical fracture, and hydrogen bonding during the steam explosion process [85]. Zhai et al. [80] investigated the effect of steam explosion on *Rosa roxburghii* pomace DF using SEM (Figure 3). The surface of *R. roxburghii* pomace insoluble dietary fiber (IDF) before steam explosion modification (0-IDF) was dense, regular, flat, and smooth without voids. After steam explosion modification (SE-IDF), the surface of *R. roxburghii* pomace IDF presented a large number of folds, and the surface structure was thin and loose with a porous honeycomb structure. The surface of *R. roxburghii* pomace SDF before steam explosion modification (0-SDF) was laminar with a relatively smooth scale-like structure, while the surface of *R. roxburghii* pomace SDF (SE-SDF) was rough and fluffy with a large number of foamy spongy structures. Wang et al. [42] reported that the surface of orange peel SDF changed from smooth to wrinkled and porous after steam explosion. Li et al. [66] investigated the effect of steam explosion on the apparent morphology of tartary buckwheat bran and its free polyphenol extracts. The surface morphology of tartary buckwheat bran without steam explosion was dense and intact, with small porosity and little detached debris. After steam explosion, the surface was separated and disrupted. Compared with the steam-explosion-treated free polyphenol extracts, the untreated free polyphenols had small round-like particle structure with more uniform distribution.

##### Effect of Steam Explosion on the Components of Bioactive Components

Zhang et al. [51] used FT-IR to investigate the effect of steam explosion on the structural composition of *Camellia* seed cake protein and found that there was no difference in the structural and chemical compositions between the natural and steam-explosion-treated samples, which suggests that steam explosion does not generate new functional groups. Qin et al. [58] concluded that steam explosion does not destroy the main components of bovine bone. Liu et al. [86], who investigated the effect of steam explosion on the composition of *Ampelopsis grossedentata* polysaccharides by monosaccharide composition, FR-IR, and NMR analyses, reported that steam explosion did not significantly affect the monosaccharide composition of the polysaccharides, nor did it change the major functional groups or the major conformations of the polysaccharides. Tanpichai et al. [87] analyzed pineapple leaf fiber using X-ray diffraction isothermal and found that steam explosion did not change the crystal structure of pineapple leaf fiber, but it caused changes in its crystallinity.

In summary, steam explosion causes significant changes in the morphology of the bioactive components. These changes are manifested as the surfaces of the bioactive components become rough and porous, exposing the internal structures. However, steam explosion does not change the composition of the bioactive components.

#### 5.3.2. Effect of Steam Explosion on Functional Properties of Bioactive Constituents

The effect of steam explosion on the structural properties of bioactive ingredients leads to changes in their functional properties. Wang et al. [8] concluded that reduced particle size and increased surface area favor the in vitro binding ability of okra. Cheng et al. [28] treated adzuki beans with steam explosion and found that steam explosion caused adzuki beans to form larger cavities and cell gaps, which contributed to the release of polyphenols and to improved antioxidant capacity. Similar findings were revealed by Li et al. [66]. Shen et al. [88] found that steam explosion improved the cholesterol adsorption capacity of black soybean hull SDF. However, excessive steam explosion reduced the biological activity of the active ingredients. Cui et al. [63] found that excessive steam explosion decreased the oil-holding capacity of grape pomace IDF.

In summary, steam explosion changes the structure of the material and increases the efficiency of solvent extraction, which is more favorable to the extraction of the target product. Moderate steam explosion can improve the quality of the target product.

## 6. Conclusions

Steam explosion technology is widely used in the pretreatment of food processing by-products because of its efficiency, environmental protection, and low cost. Steam explosion has adequate wall-breaking ability, which improves the extraction rate and biological activity of active ingredients. Consequently, steam explosion is one of the most promising pretreatment methods.

However, the available research on steam explosion technology is limited. The mechanism of action has not been fully elucidated. Additionally, the effects of the technology on the nutritional composition, functional components, and physicochemical properties of raw materials need to be further explored. For the development of steam explosion technology in the food sector, we need to (1) evaluate the physical and chemical changes of the active ingredients in the material during steam explosion; (2) identify a method to recover the steam waste heat after blasting and reuse it to reduce the energy loss; (3) develop systematic innovations to generate more efficient steam explosion equipment, given that the degree of automation is low and the loud explosion in the explosive decompression process generates noise pollution; (4) develop a standard for the use of steam explosion based on the characteristics of different food processing by-products; and (5) explore potential applications of steam explosion technology in areas other than food processing. Finally, we should further analyze the mechanism of steam explosion technology, study the factors and internal laws that affect the effect of steam explosion, break through the technical bottleneck of biomass conversion for greater efficacy, and foment the application of steam explosion technology.

## Figures and Tables

**Figure 1 foods-12-03307-f001:**
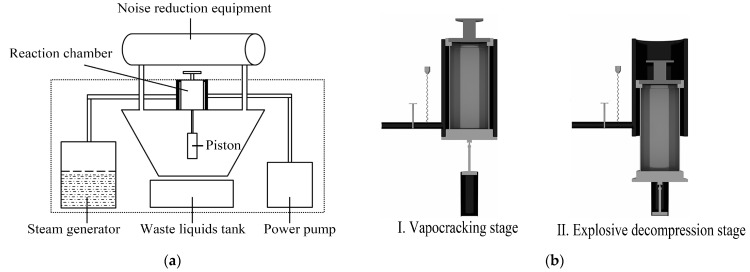
(**a**) Installation diagram of steam explosion equipment. (**b**) Working principle diagram of the steam explosion piston system.

**Figure 2 foods-12-03307-f002:**
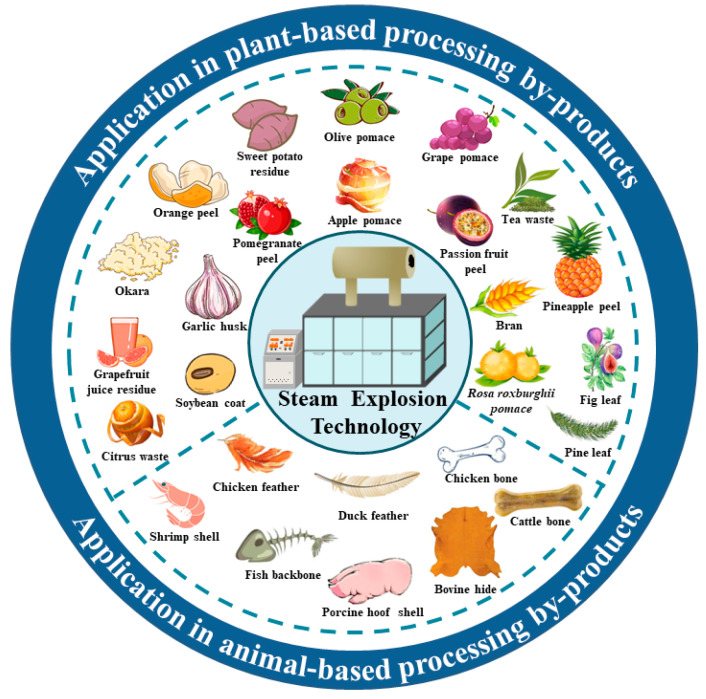
Application of steam explosion technology in food processing by-products.

**Figure 3 foods-12-03307-f003:**
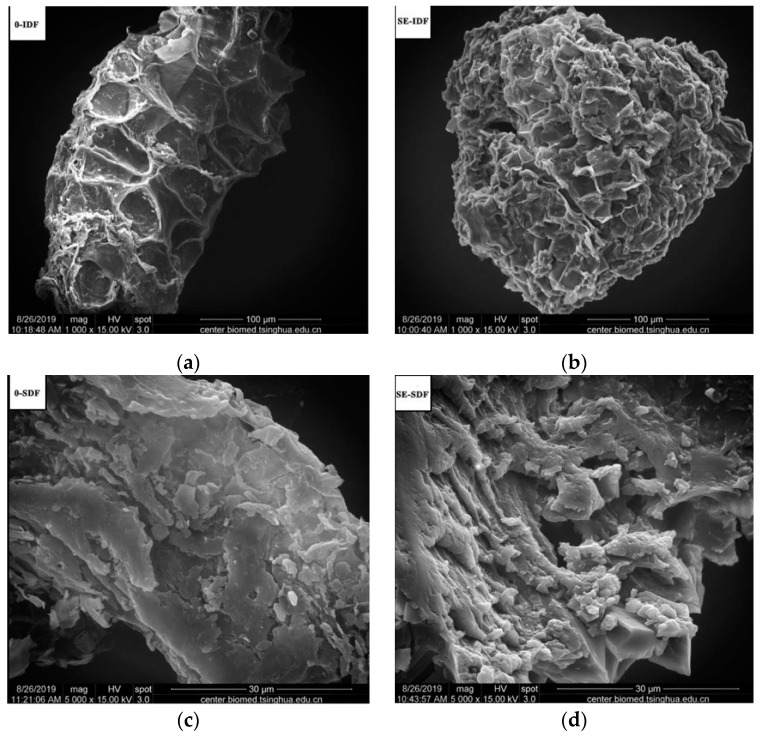
(**a**) 0-IDF without steam explosion treatment; (**b**) SE-IDF with steam explosion treatment; (**c**) 0-SDF without steam explosion treatment; (**d**) SE-SDF with steam explosion treatment [80].

**Table 1 foods-12-03307-t001:** The difference between steam explosion technology and traditional screw extrusion technology.

	Steam Explosion Technology	Traditional Screw Extrusion Technology
Duration of pressure release	Short, instant pressure relief	Long
Sound of pressure release	Loud explosion	Noiseless
Adjustable parameters	Steam explosion pressure and residence time	Extrusion pressure, temperature, and screw speed
Sources of pressure	Water steam	Screw spiral extrusion space
Source of temperature	Water steam	Frictional heat generation or external heating
Power conversion efficiency	High, near adiabatic expansion	Low
Material suitability	High	Low, strict requirements for the components of the material
Water content at discharge	High	Low
Material temperature at discharge	Lower, cooling down obviously after discharge	High, requiring longer cooling time
Material form at discharge	Slightly changed	Extrusion into strips, increased density and hardness
Material uniformity at discharge	High	Low
Equipment loss	Low	High, need to replace the screw regularly

**Table 3 foods-12-03307-t003:** Application of steam explosion technology in plant-based processing by-products.

By-Products	Extract	Steam Explosion Conditions	Results	References
Pomegranate peel	Polyphenol	Steam explosion pressure: 2.0 Mpa Residence time: 120 s.	The polyphenol extraction content and extraction rate were significantly increased; the DPPH radical scavenging and total reducing power reached the strongest level.	[65]
Tartary buckwheat bran	Polyphenol	Steam explosion pressure: 1.6 Mpa Residence time: 60 s.	Decrease in free polyphenol content and increase in bound polyphenol content. Total polyphenol scavenging activity of organic free radicals DPPH and oxygen radical absorption capacity were significantly increased.	[66]
Olive pomace	Polyphenol	Steam explosion temperature: 200 °C.	Increased total phenolic content.	[67]
Barley bran	Polyphenol	Steam explosion temperature: 220 °C Residence time: 120 s.	The total soluble phenol content was 9 times higher than that of untreated samples, and the antioxidant capacity was enhanced.	[68]
Wheat bran	Polyphenol	Steam explosion pressure: 2.5 Mpa Residence time: 30 s.	Release of bound phenolic compounds to enhance the antioxidant activity and antiproliferative activity of wheat bran.	[69]
Soybean coat	Polyphenol	Steam explosion pressure: 1.5 Mpa Residence time: 90 s.	Promotes the release of phenolic substances, which enhances antioxidant activity and nutritional value.	[13]
Garlic husk	Polyphenol	Steam explosion temperature: 200 °C Residence time: 300 s.	Increased free radical scavenging activity and a significant increase in the number of phenolic compounds.	[70]
Tea waste	Polyphenols, caffeine, saponin, and water-soluble sugars	Steam explosion pressure: 0.2, 0.4, 0.6, 0.8, and 1.0 MPa Residence time: 180 s	The solubility of tea waste was increased by 22.4%, and the extraction efficiency was improved. Under different steam explosion treatment conditions, the contents of tea polyphenols, caffeine, saponins, and water-soluble sugars were maximized by 15.5%, 14.1%, 28.8%, and 74.8%, respectively.	[71]
Grapefruit juice residue	Pectic hydrocolloids, sugars, peel oil, flavonoids, and phenolics	Steam explosion temperature: 170 °C Residence time: 480 s.	A large amount of pectic hydrocolloids, sugars, peel oil, flavonoids, and phenolics can be extracted	[72]
Grape pomace	Phenolics, flavonoids, and dietary fiber	Steam explosion pressure: 0.4, 0.8, and 1.2 MPa Residence time: 60 and 180 s.	Increased production of free phenolics and free flavonoids. Enhanced the antioxidant activity of the free extract and reduced the activity of the bound extract. Increased the yield of soluble dietary fiber and improved its physicochemical properties	[63]
Fig leaf	Flavonoids	Steam explosion pressure: 0.2 MPa Residence time: 180 s.	Flavonoid extraction rate increased by 55.9%.	[73]
Pine (*Larix olgensis Henry*)	Flavonoids	Steam explosion pressure: 1.5 MPa Residence time: 60 s.	The flavonoids extracted reached 50.8 rutin equivalents mg/g dry weight, which was 2.54-fold that of the untreated sample.	[74]
Citrus sinensis juice processing waste	Sugars, pectic hydrocolloids, flavonoids, and peel oil	Steam explosion temperature: 130, 150, and 170 °C Residence time: 60, 120, 240, and 480 s.	Increased amount of glucose or fructose, less peel oil in the raw material, and elevated flavonoid and pectin content.	[75]
Passion fruit peel	Pectin	Steam explosion pressure: 0.6 MPa Residence time: 120 s.	The extraction rate of pectin increased; the degree of esterification and molecular weight decreased.	[76]
Lime pectin peel	Pectin	Steam explosion temperature: 120, 130, 140, and 150 °C Residence time: 60, 120, and 180 s.	Recovery of most major pectic sugars increased, and change in rheological properties.	[77]
Tartary buckwheat bran	Dietary fiber	Steam explosion pressure: 1.2 MPa Residence time: 90 s.	Decreased the levels of fasting blood glucose and glycosylated hemoglobin while improved oral glucose tolerance; insulin resistance; and injuries of liver, pancreas, and colon in diabetic db/db mice.	[78]
Pineapple peel	Dietary fiber	Steam explosion pressure: 1.5 MPa Residence time: 30 s.	The content of soluble dietary fiber was increased, and the antioxidant capacity and antioxidant activity were also improved.	[79]
*Rosa roxburghii pomace*	Dietary fiber	Steam explosion pressure: 0.87 MPa Residence time: 97 s.	The content of insoluble dietary fiber decreased, and the content of soluble dietary fiber increased. The insoluble dietary fiber and soluble dietary fiber structures were disrupted, and the specific surface area increased. The thermal stability of soluble dietary fiber was increased. The water and capacity of insoluble dietary fiber was improved, and the oil-holding power of insoluble dietary fiber and soluble dietary fiber was significantly increased.	[80]
Wheat bran	Soluble dietary fiber	Steam explosion pressure: 0.5, 0.8, and 1.2 MPa Residence time: 180 and 300 s.	Facilitating the dissolution of soluble dietary fibers and promoting its hydration properties. Enhances the elasticity of wheat dough.	[81]
Orange peel	Soluble dietary fiber	Steam explosion pressure: 0.4, 0.6, 0.8, 1.0, and 1.2 MPa Residence time: 420 s.	Steam explosion coupled with sulfuric-acid soaking pretreatment. Increased SDF content and high water solubility, water-holding capacity, oil-holding capacity, swelling capacity, emulsifying activity, emulsion stability, and foam stability. Furthermore, exhibited significantly higher binding capacity for three toxic cations (Pb, As, and Cu), as well as better heat resistance.	[42]
Buckwheat bran	Soluble dietary fiber	Steam explosion pressure: 0.8 MPa Residence time: 300 s.	Soluble dietary fiber content increased from 6.1% to 11.2%, and its physicochemical properties and antioxidant capacity were also improved.	[82]
Sweet potato residue	Soluble dietary fiber	Steam explosion pressure: 0.35 MPa Residence time: 121 s.	The content of soluble dietary fiber increased by 18.78% compared to the untreated group. The water-holding capacity, oil-holding capacity, and swelling capacity of soluble dietary fiber were enhanced and became poriferous, loose, and dilatant.	[33]
Apple pomace	Soluble dietary fiber	Steam explosion pressure: 0.51 MPa Residence time: 168 s.	The yield of soluble dietary fiber was increased to 29.85%, which was 4.76 times higher than the control group. Steam explosion altered the morphology of soluble dietary fiber and exhibited higher functionality values.	[46]
Soybean hull	Soluble dietary fiber	Steam explosion pressure: 0.6, 1.2, 1.8, and 2.4 MPa Residence time: 60, 120, and 180 s.	Improved the content of soluble dietary fiber and in vitro gross energy digestibility, and reducing sugar yield improved by 55.24%.	[83]
Okara	Soluble dietary fiber	Steam explosion pressure: 1.5 MPa Residence time: 30 s.	Soluble dietary fiber content increased from 1.34% to 36.28%, an increase of 26 times, increasing 26 times compared to the control. Soluble dietary fiber content increased from 1.34% to 36.28%. Water solubility of okra increased significantly. And depolymerized proteins, leading to the increase in low molecular proteins.	[84]

## Data Availability

The datasets generated for this study are available on request to the corresponding author.

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
