# Peer review of "The Principle of Steam Explosion Technology and Its Application in Food Processing By-Products"

_foods, 2023, doi:10.3390/foods12173307_

Round 1

Reviewer 1 Report

The manuscript clearly explains all the aspects of the study background. It is in good systematic. The manuscript is technically sound and possesses excellent presentation clarity. A few comments regarding the manuscript as follows:

Equation: subscripts should be correct in the explanations.

The unit of each equation should be added.

An English revision should be carried out since the manuscript has some statements and minor grammar mistakes that must be clarified. 

Author Response

Cover letter

Thank you very much for your E-mail to report the status of our manuscript entitled “Principle of Steam Explosion Technology and its Application in Food Processing By-products” (FOODS-2587374) again, which is submitted to “FOODS” to consider for publication. According to the reviewers’ comments, we have revised carefully with colored text (Red), and detailed corrections are listed below point by point:

The manuscript clearly explains all the aspects of the study background. It is in good systematic. The manuscript is technically sound and possesses excellent presentation clarity. A few comments regarding the manuscript as follows:

Thanks for the referee’s kind advice. We have revised our manuscript carefully and listed corrections point by point.

Point 1: Equation: subscripts should be correct in the explanations.

Response 1: Comments have been taken into account. The manuscript was revised accordingly. (Page 4, line 159).

Point 2: The unit of each equation should be added.

Response 2: Comments have been taken into account. The manuscript was revised accordingly. (Page 5, equation 2).

Point 3: An English revision should be carried out since the manuscript has some statements and minor grammar mistakes that must be clarified.

Response 3: Thanks for the reviewer’s comments. We have made English revisions to the manuscript.

This manuscript has not been published elsewhere and is not under consideration by another journal. Each of the authors has approved the final version of the manuscript, agree with this submission to FOODS, and report no conflicts of interest. Also, we have sent this manuscript to International Science Editing (http://www.internationalscienceediting.com) for further edit, and revised the grammar errors and inappropriate long sentences accordingly.

On behalf of the authors, I look forward to hearing from you at your earliest convenience.

Sincerely,

Thanks very much for your attention and consideration.

Sincerely yours,

Zebin Guo

Reviewer 2 Report

The paper is a well-written and informative review of the steam explosion technology. The authors start by providing an overview of the technology, its similarities and differences with traditional screw extrusion technology, and the factors that affect the process. They then review the applications of steam explosion technology in food processing by-products in recent years. Finally, they provide an outlook on the development of steam explosion technology and its potential applications in the food processing field. The study is interesting. However, before the final acceptance can be made. I would like the authors to address the following points:

The paper only reviews the applications of steam explosion technology in food processing by-products in recent years. More research is needed to explore the potential applications of this technology in other areas.

The paper does not discuss the safety of steam explosion technology. More research is needed to ensure that this technology is safe for food processing.

The paper does not provide any economic analysis of steam explosion technology.

Overall, the paper is a valuable contribution to the literature on steam explosion technology. 

Page 1 Line 9: "has shown great promise in food processing" should be "has shown great promise for food processing"

Page 1 Line 38: "stands out because of its obvious advantages" should be "stands out due to its obvious advantages"

Page 2 Line 78: "reactor inner cavity" should be "reactor's inner cavity"

Author Response

Cover letter

Thank you very much for your E-mail to report the status of our manuscript entitled “Principle of Steam Explosion Technology and its Application in Food Processing By-products” (FOODS-2587374) again, which is submitted to “FOODS” to consider for publication. According to the reviewers’ comments, we have revised carefully with colored text (Red), and detailed corrections are listed below point by point:

Reviewer 2:

The paper is a well-written and informative review of the steam explosion technology. The authors start by providing an overview of the technology, its similarities and differences with traditional screw extrusion technology, and the factors that affect the process. They then review the applications of steam explosion technology in food processing by-products in recent years. Finally, they provide an outlook on the development of steam explosion technology and its potential applications in the food processing field. The study is interesting. However, before the final acceptance can be made. I would like the authors to address the following points:

Thanks for the referee’s kind advice. We have revised our manuscript carefully and listed corrections point by point.

Point 1: The paper only reviews the applications of steam explosion technology in food processing by-products in recent years. More research is needed to explore the potential applications of this technology in other areas.

Response 1: Comments have been taken into account. Some necessary information and statements have been supplemented accordingly.

.Steam explosion is no longer limited to areas such as pulping and wood processing [1], animal feed processing [2] and bioenergy production [3], but to food processing as well. (Page 2, line 54 - 56)

  1. Martin-Sampedro, R.; Eugenio, M.E.; Moreno, J.A.; Revilla, E.; Villar, J.C. Integration of a Kraft Pulping Mill into a Forest Biorefinery: Pre-Extraction of Hemicellulose by Steam Explosion versus Steam Treatment. Bioresour. Technol. 2014, 153, 236–244, doi:10.1016/j.biortech.2013.11.088.
  2. Xie, H.; Li, Z.; Wang, Z.; Mao, G.; Zhang, H.; Wang, F.; Chen, H.; Yang, S.; Tsang, Y.F.; Lam, S.S.; et al. Instant Catapult Steam Explosion: A Rapid Technique for Detoxification of Aflatoxin-Contaminated Biomass for Sustainable Utilization as Animal Feed. J. Clean. Prod. 2020, 255, 120010, doi:10.1016/j.jclepro.2020.120010.
  3. Hoang, A.T.; Nguyen, X.P.; Duong, X.Q.; Agbulut, U.; Len, C.; Nguyen, P.Q.P.; Kchaou, M.; Chen, W.-H. Steam Explosion as Sustainable Biomass Pretreatment Technique for Biofuel Production: Characteristics and Challenges. Bioresour. Technol. 2023, 385, 129398, doi:10.1016/j.biortech.2023.129398.

(5) Explore potential applications of steam explosion technology in areas other than food processing. (Page 11, line 336 - 337).

Point 2: The paper does not discuss the safety of steam explosion technology. More research is needed to ensure that this technology is safe for food processing.

Response 2: Comments have been taken into account. The manuscript was revised accordingly.

Steam explosion technology uses water vapor to treat the raw material and therefore does not produce any toxic or hazardous substances [1], and at the same time, because of its excellent processing effect and applicability, it has been widely researched in the field of food processing, including wall-breaking treatment of raw materials, extraction of bioactive components, and improvement of functional properties of bioactive components. (Page 5 - 6, line 207 - 212).

  1. 1. Chen, Y.; Shan, S.; Cao, D.; Tang, D. Steam Flash Explosion Pretreatment Enhances Soybean Seed Coat Phenolic Profiles and Antioxidant Activity. FOOD Chem. 2020, 319, 126552, doi:10.1016/j.foodchem.2020.126552.

Point 3: The paper does not provide any economic analysis of steam explosion technology.

Response 3: Comments have been taken into account. The manuscript was revised accordingly.

Fourth, Significant economic advantages: the traditional screw extrusion relies on the screw to process the material, which causes high wear and tear on the screw and requires regular replacement [1]. At the same time, energy consumption is a major part of the operating costs of steam explosion equipment, while steam explosion technology has a very high energy conversion efficiency (thermal energy to mechanical energy), so that the actual energy consumption is at a very low level [2]. (Page 3 - 4, line 131 - 136).

  1. 1. Sun, X.; Yu, C.; Fu, M.; Wu, D.; Gao, C.; Feng, X.; Cheng, W.; Shen, X.; Tang, X. Extruded Whole Buckwheat Noodles: Effects of Processing Variables on the Degree of Starch Gelatinization, Changes of Nutritional Components, Cooking Characteristics and in Vitro Starch Digestibility. FOOD Funct. 2019, 10, 6362–6373, doi:10.1039/c9fo01111k.
  2. 2. Yu, Z.; Zhang, B.; Yu, F.; Xu, G.; Song, A. A Real Explosion: The Requirement of Steam Explosion Pretreatment. Technol. 2012, 121, 335–341, doi:10.1016/j.biortech.2012.06.055.

Point 4: Page 1 Line 9: "has shown great promise in food processing" should be "has shown great promise for food processing".

Response 4: Comments have been taken into account. The manuscript was revised accordingly. (Page 1, line 10).

Point 5: Page 1 Line 38: "stands out because of its obvious advantages" should be "stands out due to its obvious advantages".

Response 5: Comments have been taken into account. The manuscript was revised accordingly. (Page 2, line 52).

Point 6: Page 2 Line 78: "reactor inner cavity" should be "reactor's inner cavity". Response 6: Comments have been taken into account. The manuscript was revised accordingly. (Page 2, line 93).

This manuscript has not been published elsewhere and is not under consideration by another journal. Each of the authors has approved the final version of the manuscript, agree with this submission to FOODS, and report no conflicts of interest. Also, we have sent this manuscript to International Science Editing (http://www.internationalscienceediting.com) for further edit, and revised the grammar errors and inappropriate long sentences accordingly.

On behalf of the authors, I look forward to hearing from you at your earliest convenience.

Sincerely,

Thanks very much for your attention and consideration.

Sincerely yours,

Zebin Guo

Reviewer 3 Report

The authors have submitted a manuscript in which they summarize the steam explosion technology's fundamentals and uses, examining the limitations of the technology and its development prospects in the food industry, in order to provide some guidance for future applications in the by-products of food processing.

The topic of this paper is suitable for the scope of the Journal. The novelty could be explained better. Is this the first time that a comprehensive review is written on this topic? Are there similar papers in the literature? What do this review add?

Abstract can be improved, in particular by adding a short reference to the main applications analyzed regarding food processing.

The Introduction is quite well presented. I would add a paragraph on the importance of pre-treatment methods, with a more detailed explanation of the most used methods, their strengths and weaknesses, motivating why steam explosion can help the sector.

Paragraph 3 can be improved adding a section analysing the different impact/efficiency in pre-treating food by-products matrices with examples taken from literature. Moreover, a short economic analysis of the energetic and equipment costs could be helpful.

Paragraph 4.3 is not well explained. Please, rewrite it and add some details as this is a crucial factor affecting the efficiency of the pretreatment.

Sections 5.1 and 5.2 are just a list of examples. It would be useful to go into more detail on the discussion by highlighting the advantages of the method on the pre-treatment of the matrix: e.g., which components are specifically broken down in the various examples mentioned (e.g. proteins, cellulose, hemicellulose)?

In paragraphs 5.3 briefly discuss the reasons for which the mentioned effects occur

Good english. Some Sections are not well presented (e.g., 4.3) and should be revised

Author Response

Cover letter

Thank you very much for your E-mail to report the status of our manuscript entitled “Principle of Steam Explosion Technology and its Application in Food Processing By-products” (FOODS-2587374) again, which is submitted to “FOODS” to consider for publication. According to the reviewers’ comments, we have revised carefully with colored text (Red), and detailed corrections are listed below point by point:

Reviewer 3:

The authors have submitted a manuscript in which they summarize the steam explosion technology's fundamentals and uses, examining the limitations of the technology and its development prospects in the food industry, in order to provide some guidance for future applications in the by-products of food processing.

Thanks for the referee’s kind advice. We have revised our manuscript carefully and listed corrections point by point.

Point 1: The topic of this paper is suitable for the scope of the Journal. The novelty could be explained better. Is this the first time that a comprehensive review is written on this topic? Are there similar papers in the literature? What do this review add?

Response 1: Thanks for the reviewer’s comments. Similar papers in the literature are as follows:

  1. 1. Ma, C.; Ni, L.; Guo, Z.; Zeng, H.; Wu, M.; Zhang, M.; Zheng, B. Principle and Application of Steam Explosion Technology in Modification of Food Fiber. Foods 2022, 11, 3370, doi:10.3390/foods11213370.
  2. 2. Sui, W.; Chen, H. Multi-Stage Energy Analysis of Steam Explosion Process. Eng. Sci. 2014, 116, 254–262, doi:10.1016/j.ces.2014.05.012.
  3. 3. Yu, Z.; Zhang, B.; Yu, F.; Xu, G.; Song, A. A Real Explosion: The Requirement of Steam Explosion Pretreatment. Bioresour. Technol. 2012, 121, 335–341, doi:10.1016/j.biortech.2012.06.055.

Existing literature only introduced the principle of steam explosion or the effect on a certain active substance. This paper, based on existing literature, comprehensively introduces the steam explosion technology and integrates its application in food processing by-products, aiming to expand the application areas of steam explosion technology.

Point 2: Abstract can be improved, in particular by adding a short reference to the main applications analyzed regarding food processing.

Response 2: Comments have been taken into account. The manuscript was revised accordingly.

The results of the current study indicate that moderate steam explosion treatment can improve the quality and extraction rate of the target products. (Page 1, line 14 - 15).

Point 3: The Introduction is quite well presented. I would add a paragraph on the importance of pre-treatment methods, with a more detailed explanation of the most used methods, their strengths and weaknesses, motivating why steam explosion can help the sector.

Response 3: Comments have been taken into account. The manuscript was revised accordingly.

Currently, common food pretreatment methods include physical, chemical, and biological methods [1]. The chemical method mainly destroys the structure of raw materials by adding acid or alkali to achieve the purpose of pretreatment [2]. The chemical method has the advantage of short treatment time, but there are problems such as waste liquid pollution of the environment and the possible introduction of toxic and harmful chemical groups during the treatment process [3], so there are certain risks when applied to food processing. Biological methods mainly use enzymes or microorganisms to enzymatically dissolve or ferment raw materials, thus completing the pretreatment of raw materials [4], the reaction of the method is mild, but the reaction conditions are extremely harsh and expensive, and it is difficult to realize the industrialization of the application [5]. Compared to chemical and biological methods, physical methods have received extensive attention from scholars due to their advantages such as high efficiency, low cost, and absence of pollutants [6].

Steam explosion technology stands out among many physical pretreatment methods due to its obvious advantages in energy consumption, and treatment efficiency [7]. (Page 1 - 2, line 38 - 53).

  1. 1. Wang, Q.; Shen, P.; Chen, B. Ultracentrifugal Milling and Steam Heating Pretreatment Improves Structural Characteristics, Functional Properties, and in Vitro Binding Capacity of Cellulase Modified Soy Okara Residues. FOOD Chem. 2022, 384, 132526, doi:10.1016/j.foodchem.2022.132526.
  2. 2. Qi, J.; Li, Y.; Masamba, K.G.; Shoemaker, C.F.; Zhong, F.; Majeed, H.; Ma, J. The Effect of Chemical Treatment on the In Vitro Hypoglycemic Properties of Rice Bran Insoluble Dietary Fiber. FOOD Hydrocoll. 2016, 52, 699–706, doi:10.1016/j.foodhyd.2015.08.008.
  3. 3. Huang, S.; He, Y.; Zou, Y.; Liu, Z. Modification of Insoluble Dietary Fibres in Soya Bean Okara and Their Physicochemical Properties. J. FOOD Sci. Technol. 2015, 50, 2606–2613, doi:10.1111/ijfs.12929.
  4. 4. Wen, Y.; Niu, M.; Zhang, B.; Zhao, S.; Xiong, S. Structural Characteristics and Functional Properties of Rice Bran Dietary Fiber Modified by Enzymatic and Enzyme-Micronization Treatments. LWT-FOOD Sci. Technol. 2017, 75, 344–351, doi:10.1016/j.lwt.2016.09.012.
  5. 5. Gan, J.; Xie, L.; Peng, G.; Xie, J.; Chen, Y.; Yu, Q. Systematic Review on Modification Methods of Dietary Fiber. FOOD Hydrocoll. 2021, 119, 106872, doi:10.1016/j.foodhyd.2021.106872.
  6. 6. Sarker, T.R.; Pattnaik, F.; Nanda, S.; Dalai, A.K.; Meda, V.; Naik, S. Hydrothermal Pretreatment Technologies for Lignocellulosic Biomass: A Review of Steam Explosion and Subcritical Water Hydrolysis. CHEMOSPHERE 2021, 284, 131372, doi:10.1016/j.chemosphere.2021.131372.
  7. Qi, J.; Li, Y.; Masamba, K.G.; Shoemaker, C.F.; Zhong, F.; Majeed, H.; Ma, J. The Effect of Chemical Treatment on the In Vitro Hypoglycemic Properties of Rice Bran Insoluble Dietary Fiber. FOOD Hydrocoll. 2016, 52, 699–706, doi:10.1016/j.foodhyd.2015.08.008.

Point 4: Paragraph 3 can be improved adding a section analysing the different impact/efficiency in pre-treating food by-products matrices with examples taken from literature. Moreover, a short economic analysis of the energetic and equipment costs could be helpful.

Response 4: Comments have been taken into account. The manuscript was revised accordingly.

3.1. Effect of Steam explosion and traditional screw extrusion on material treatment

Steam explosion has similar effects to traditional conventional screw extrusion. Both technologies modify the material in a way that minimizes food degradation and destroys anti-nutrients [1]. In addition, the microstructure and physicochemical properties of the material can be modified by steam explosion and traditional screw extrusion with improvements in the functional properties [2]. Wang et al. [3] and Qiao et al. [4], who subjected sweet potato residue to steam explosion and screw extrusion, respectively, reported that sweet potato residue SDF become more loose and porous, that the extracted amount of SDF increased, and that the water-holding capacity, oil-holding capacity, and swelling capacity improved to different degrees. Furthermore, Wang et al. [5] subjected wheat bran to extrusion and found that wheat bran SDF was enhanced along with its antioxidant capacity, similar to the findings of Feng et al. [6], who subjected wheat bran to steam explosion. (Page 3, line 111 - 123).

  1. 1. Li, C.; Huang, X.; Xi, J. Steam Explosion Pretreatment to Enhance Extraction of Active Ingredients: Current Progress and Future Prospects. Rev. FOOD Sci. Nutr. 2023, doi:10.1080/10408398.2023.2181760.
  2. 2. Cheng, A.; Hou, C.; Sun, J.; Wan, F. Effect of Steam Explosion on Phenolic Compounds and Antioxidant Capacity in Adzuki Beans. Sci. FOOD Agric. 2020, 100, 4495–4503, doi:10.1002/jsfa.10490.
  3. 3. Uitterhaegen, E.; Evon, P. Twin-Screw Extrusion Technology for Vegetable Oil Extraction: A Review. FOOD Eng. 2017, 212, 190–200, doi:10.1016/j.jfoodeng.2017.06.006.
  4. 4. Zhou, C.; Wu, M.; Sun, D.; Wei, W.; Yu, H.; Zhang, T. Twin-Screw Extrusion of Oat: Evolutions of Rheological Behavior, Thermal Properties and Structures of Extruded Oat in Different Extrusion Zones. FOODS 2022, 11, 2206, doi:10.3390/foods11152206.
  5. 5. Sun, X.; Yu, C.; Fu, M.; Wu, D.; Gao, C.; Feng, X.; Cheng, W.; Shen, X.; Tang, X. Extruded Whole Buckwheat Noodles: Effects of Processing Variables on the Degree of Starch Gelatinization, Changes of Nutritional Components, Cooking Characteristics and in Vitro Starch Digestibility. FOOD Funct. 2019, 10, 6362–6373, doi:10.1039/c9fo01111k.
  6. 6. He, H.; Zhang, X.; Liao, W.; Shen, J. Characterization and in Vitro Digestion of Rice Starch/Konjac Glucomannan Complex Prepared by Screw Extrusion and Its Impact on Gut Microbiota. FOOD Hydrocoll. 2023, 135, 108156, doi:10.1016/j.foodhyd.2022.108156.

Fourth, Significant economic advantages: the traditional screw extrusion relies on the screw to process the material, which causes high wear and tear on the screw and requires regular replacement [1]. At the same time, energy consumption is a major part of the operating costs of steam explosion equipment, while steam explosion technology has a very high energy conversion efficiency (thermal energy to mechanical energy), so that the actual energy consumption is at a very low level [3]. (Page 3 - 4, line 131 - 136).

  1. 1. Li, C.; Huang, X.; Xi, J. Steam Explosion Pretreatment to Enhance Extraction of Active Ingredients: Current Progress and Future Prospects. Rev. FOOD Sci. Nutr. 2023, doi:10.1080/10408398.2023.2181760.
  2. 2. Yu, Z.; Zhang, B.; Yu, F.; Xu, G.; Song, A. A Real Explosion: The Requirement of Steam Explosion Pretreatment. Technol. 2012, 121, 335–341, doi:10.1016/j.biortech.2012.06.055.

Point 5: Paragraph 4.3 is not well explained. Please, rewrite it and add some details as this is a crucial factor affecting the efficiency of the pretreatment.

Response 5: Comments have been taken into account. The manuscript was revised accordingly.

This is because the raw material particle size becomes small, the surface area involved in the steam explosion reaction increases, so the treatment is enhanced. However, too small a particle size in the steam explosion treatment is prone to agglomeration, which prevents the further penetration of saturated steam, which in turn makes the steam explosion reaction is not sufficient [1]. (Page 5, line 200 - 204).

  1. 1. Sui, W.; Chen, H. Multi-Stage Energy Analysis of Steam Explosion Process. Eng. Sci. 2014, 116, 254–262, doi:10.1016/j.ces.2014.05.012.

Point 6: Sections 5.1 and 5.2 are just a list of examples. It would be useful to go into more detail on the discussion by highlighting the advantages of the method on the pre-treatment of the matrix: e.g., which components are specifically broken down in the various examples mentioned (e.g. proteins, cellulose, hemicellulose)?

Response 6: Comments have been taken into account. The manuscript was revised accordingly.

The reason for the increased protein extraction rate by steam explosion treatment may be that steam explosion alters the subunits of proteins, leading to depolymerization and a wider distribution of molecular weights, resulting in the formation of non-disulfide bonds between protein molecules [1]. The reason why steam explosion treatment improves the solubility of denatured proteins is that during the steam explosion process, hydrophilic groups interact with proteins, leading to Maillard reactions [2]. In addition, high solubility proteins indicate the ability to release more natural peptides [3], which means that steam explosion treatment may improve certain functional properties of proteins [4]. (Page 6 - 7, line 234 - 242).

  1. 1. Zhang, Y.; Yang, R.; Zhao, W.; Hua, X.; Zhang, W. Application of High Density Steam Flash-Explosion in Protein Extraction of Soybean Meal. FOOD Eng. 2013, 116, 430–435, doi:10.1016/j.jfoodeng.2012.12.006.
  2. 2. Zhang, S.; Zheng, L.; Zheng, X.; Ai, B.; Yang, Y.; Pan, Y.; Sheng, Z. Effect of Steam Explosion Treatments on the Functional Properties and Structure of Camellia (Camellia Oleifera Abel.) Seed Cake Protein. Food Hydrocoll. 2019, 93, 189–197, doi:10.1016/j.foodhyd.2019.02.017.
  3. 3. Malomo, S.A.; He, R.; Aluko, R.E. Structural and Functional Properties of Hemp Seed Protein Products. FOOD Sci. 2014, 79, C1512–C1521, doi:10.1111/1750-3841.12537.
  4. 4. Ma, M.; Ren, Y.; Xie, W.; Zhou, D.; Tang, S.; Kuang, M.; Wang, Y.; Du, S.-K. Physicochemical and Functional Properties of Protein Isolate Obtained from Cottonseed Meal. FOOD Chem. 2018, 240, 856–862, doi:10.1016/j.foodchem.2017.08.030.

Also, steam explosion promotes the degradation of polymers into biologically active substances. For example, the reason for the increased SDF extraction rate after steam blasting treatment is that steam blasting destroys the structure of the feedstock, degradation of cellulose and hemicellulose occurs, thus promoting SDF release and extraction [1]. (Page 7, line 252 - 256).

  1. 1. Li, W.; Zhang, X.; He, X.; Li, F.; Zhao, J.; Yin, R.; Ming, J. Effects of Steam Explosion Pretreatment on the Composition and Biological Activities of Tartary Buckwheat Bran Phenolics. FOOD Funct. 2020, 11, 4648–4658, doi:10.1039/d0fo00493f.

Point 7: In paragraphs 5.3 briefly discuss the reasons for which the mentioned effects occur.

Response 7: This phenomenon occurs as a result of acid hydrolysis, thermal degradation, mechanical fracture and hydrogen bonding during the steam explosion process [1]. (Page 9, line 264 - 266).

  1. 1. Ma, C.; Ni, L.; Guo, Z.; Zeng, H.; Wu, M.; Zhang, M.; Zheng, B. Principle and Application of Steam Explosion Technology in Modification of Food Fiber. FOODS 2022, 11, 3370, doi:10.3390/foods11213370.

The effect of steam explosion on the structural properties of bioactive ingredients leads to changes in their functional properties. (Page 11, line 305 - 306).

Point 8: Good english. Some Sections are not well presented (e.g., 4.3) and should be revised.

Response 8: Thanks for the reviewer’s comments. It has been revised accordingly.

Thank the above reviewers for their comments on this article. We have made changes according to the relevant comments accordingly in the manuscript. We believe that the findings of this study are relevant to the scope of FOODS and will be of interest to its readership. (Page 5, line 119 - 204).

This manuscript has not been published elsewhere and is not under consideration by another journal. Each of the authors has approved the final version of the manuscript, agree with this submission to FOODS, and report no conflicts of interest. Also, we have sent this manuscript to International Science Editing (http://www.internationalscienceediting.com) for further edit, and revised the grammar errors and inappropriate long sentences accordingly.

On behalf of the authors, I look forward to hearing from you at your earliest convenience.

Sincerely,

Thanks very much for your attention and consideration.

Sincerely yours,

Zebin Guo